# Artificial Extracellular Vesicles Generated from T Cells Using Different Induction Techniques

**DOI:** 10.3390/biomedicines12040919

**Published:** 2024-04-20

**Authors:** Ekaterina A. Zmievskaya, Sabir A. Mukhametshin, Irina A. Ganeeva, Elvina M. Gilyazova, Elvira T. Siraeva, Marianna P. Kutyreva, Artur A. Khannanov, Youyong Yuan, Emil R. Bulatov

**Affiliations:** 1Institute of Fundamental Medicine and Biology, Kazan Federal University, 420008 Kazan, Russia; 2A.M. Butlerov Institute of Chemistry, Kazan Federal University, 420008 Kazan, Russia; 3School of Biomedical Sciences and Engineering, South China University of Technology, Guangzhou International Campus, Guangzhou 511442, China; 4Shemyakin-Ovchinnikov Institute of Bioorganic Chemistry, Russian Academy of Sciences, 117997 Moscow, Russia

**Keywords:** T cells, extracellular vesicles, artificial vesicles, vesicle induction, ultrasonication

## Abstract

Cell therapy is at the forefront of biomedicine in oncology and regenerative medicine. However, there are still significant challenges to their wider clinical application such as limited efficacy, side effects, and logistical difficulties. One of the potential approaches that could overcome these problems is based on extracellular vesicles (EVs) as a cell-free therapy modality. One of the major obstacles in the translation of EVs into practice is their low yield of production, which is insufficient to achieve therapeutic amounts. Here, we evaluated two primary approaches of artificial vesicle induction in primary T cells and the SupT1 cell line—cytochalasin B as a chemical inducer and ultrasonication as a physical inducer. We found that both methods are capable of producing artificial vesicles, but cytochalasin B induction leads to vesicle yield compared to natural secretion, while ultrasonication leads to a three-fold increase in particle yield. Cytochalasin B induces the formation of vesicles full of cytoplasmic compartments without nuclear fraction, while ultrasonication induces the formation of particles rich in membranes and membrane-related components such as CD3 or HLAII proteins. The most effective approach for T-cell induction in terms of the number of vesicles seems to be the combination of anti-CD3/CD28 antibody activation with ultrasonication, which leads to a seven-fold yield increase in particles with a high content of functionally important proteins (CD3, granzyme B, and HLA II).

## 1. Introduction

T cell-based immunotherapy, including CAR-T, TILs, Tregs, and many others, represents one of the most promising types of modern cell therapy in the fields of oncology and autoimmune diseases. Despite its success, this type of therapy has some drawbacks—live cells are difficult to store, transport, and dose, and in the majority of cases, they have to be autologous for each patient. These challenges may limit the availability and accessibility of this therapy for patients. However, ongoing research and development are focused on overcoming these obstacles and improving the feasibility and efficacy of T cell-based immunotherapies.

One potential approach to address these issues is the use of artificial extracellular vesicles (EVs). EVs are a heterogeneous group of nano-sized lipid vesicles of cellular origin. They are typically classified into three types based on their size and mechanism of formation: exosomes, microvesicles, and apoptotic bodies. The biochemical composition and functional properties of EVs are determined by the parental cell type and physiological state, making them a cell-free analog of cell therapy. Artificial vesicles (AVs) are designed to mimic the biological properties of EVs and can be produced in a scalable and standardized manner to achieve more convenient storage, transportation, and dosage than for live cells [1].

The secretion of EVs by T cells is a natural process, particularly in response to TCR-specific activation. T cell-secreted EVs are known to carry a variety of effector proteins such as CD3, MHC-I and MHC-II, CD2, LFA-1, CXCR4, and FasL, as well as small RNAs [2]. Functionally, EVs derived from CD8^+^ T cells have been shown to possess immune-activating and antitumor properties, while EVs from Tregs have been shown to have immunomodulatory effects [1].

Overall, EVs represent a promising therapeutic agent due to their stability, biosafety, and capability to target specific tissues with better penetration than cells [3]. Some reports claim that they also have the potential to cross biological barriers such as the blood–brain barrier [4]. EVs can also be used as drug delivery vehicles and have advantages over other nanoscale delivery approaches, including better biocompatibility, longer half-life, and lower immunogenicity and toxicity. By combining the natural properties of EVs with therapeutic small molecules, the overall therapeutic efficacy can be significantly enhanced [5].

The main obstacle to EV technology becoming a viable treatment option is the difficulty of large-scale production. Natural EVs have a very low yield, sufficient for academic research but insufficient for industrial or clinical use. Up to a 500 L high-density perfusion bioreactor may need to be used to manufacture the required amount of EVs for clinical use [6].

As a result, several approaches are being investigated to increase EV production yield. The strategies include dynamic culture, hypoxia, starvation, low pH, chemical stimulation (vesiculation buffer, cytochalasin B, and sulfhydryl blocking agents), physical stimulation (shear stress, high-frequency ultrasonication (US), and irradiation), cell membrane disruption (extrusion, nitrogen cavitation, and low-frequency ultrasonication) [7].

In this study, we aimed to investigate and compare two main approaches for the generation of artificial extracellular vesicles from T cells: chemical stimulation (cytochalasin B) and cell membrane disruption (low-frequency ultrasonication). The detailed study design is presented in Figure 1.

## 2. Materials and Methods

### 2.1. T Cells’ Isolation and Expansion

Peripheral blood mononuclear cells (PBMCs) were obtained from the whole blood of a healthy donor (according to ethics approval and written informed consent) by Ficoll (Paneco, Moscow, Russia) density centrifugation at 800× *g* for 30 min. The buffy coat was collected and washed twice with Dulbecco’s phosphate-buffered saline, and the cells were seeded in 10% RPMI with L-glutamine and penicillin–streptomycin (Paneco, Moscow Russia). T cells were activated with T-cell TransAct (Miltenyi, Bergisch Gladbach, Germany) according to the manufacturer’s protocol, and IL-2 (Prospec, Rehovot, Israel) was added at a concentration of 300 IU/mL. T cells were counted every two days, and a fresh medium containing IL-2 was added to maintain a cell density of approximately 1 × 10^6^ cells/mL. After 14 days of expansion, cell populations were phenotyped by flow cytometry for CD3, CD4, and CD8 markers (Appendix A).

### 2.2. SupT1 Cell Culture and Transduction

SupT1 (ATCC CRL-1942, mycoplasma free) cells were cultured in 10% RPMI supplemented with penicillin–streptomycin and L-glutamine. To obtain AVs, the cells were washed twice with DPBS and resuspended in serum-free RPMI. To obtain SupT1(Kat+) cells, SupT1 expressing red fluorescent protein Katushka2S, lentiviral transduction was performed using second-generation lentivirus harvested from HEK293FT culture and concentrated using Amicon 15 (Merck, Darmstadt, Germany). Transduction was performed with protamine sulfate at a concentration of 50 µg/mL for 16 h. The medium was then replaced, and a selective antibiotic (blasticidine) was added after 48 h. The selection of SupT1(Kat+) continued for 1 week. The efficacy of Katushka2S transduction was determined by flow cytometry (Appendix A).

### 2.3. MV Induction and Collection

To obtain microvesicles from primary T cells, the cells were harvested on day 14, washed twice with DPBS, and seeded at a density of 1 × 10^6^ cells/mL in a serum-free AIM-V medium (Thermo Fisher, Waltham, MA, USA) containing 300 IU/mL IL-2. The cells were then divided into two parts. One portion was left in a CO_2_ incubator (ESCO, Seoul, Republic of Korea) to allow for resting T-cell culture, while the other was reactivated using T-cell TransAct (Miltenyi, Auburn, CA, USA). After 48 h of incubation, the supernatants were obtained by centrifugation to separate natural microvesicles (MVs), while the cell pellets were washed twice with DPBS, resuspended in serum-free media, and subsequently induced to generate AVs.

### 2.4. AV Induction

To obtain AVs, the cells were washed twice with DPBS and resuspended in serum-free media. AVs were then induced using either cytochalasin B or ultrasonication. For cytochalasin B induction, cytochalasin B (ChB) was added at a concentration of 10 µg/mL, and the cells were incubated in a CO_2_ incubator (ESCO, Seoul, South Korea) for 30 min. The cells were then gently vortexed for 30 s. For ultrasonication induction, the cells were subjected to ultrasonication impact using a Bandelin Sonopuls HD2200 (Bandelin, Berlin, Germany) homogenizer at cycle 5 and 20% power for 1 min.

### 2.5. MV and AV Isolation

The isolation of MVs and AVs was performed by differential centrifugation. First, the cell pellet was obtained by centrifugation at 300× *g* for 10 min, followed by centrifugation of the supernatant at 2000× *g* for 10 min to remove debris and apoptotic bodies. Finally, the vesicles were sedimented by centrifugation at 15,000× *g* for 20 min. After separation, the samples were washed twice with DPBS.

### 2.6. Nanoparticle Tracking Analysis

Nanoparticle tracking analysis (NTA) was performed using NanoSight LM-10 (Malvern Instruments, Malvern, UK). The detector was a C11440-50B CMOS camera with an FL-280 image capture sensor from Hamamatsu Photonics (Shizuoka, Japan). Measurements were performed in a special cuvette for aqueous solutions equipped with a 405 nm laser (CD version, S/N 2990491) and a sealing ring made of Kalrez material. The temperature was recorded for all measurements using an OMEGA HH804 contact thermometer (Engineering Inc., Chicago, IL, USA). Samples for analysis were collected and injected into the measuring cell using a 1 mL two-part glass syringe (tuberculin) through the Luer (Hamilton Company, Reno, NV, USA). To increase the statistical dose, the sample was pumped through the measuring chamber using a piezoelectric dispenser. Each sample was acquired six times sequentially; the acquisition time was sequential and was 60 s. The NanoSight images were processed using NTA 2.3 software applications (build 0033), as previously described [8]. The detailed work is currently presented in the “Principle of Operation” by B. Carr and A. Malloy [9]. The hydrodynamic diameter was calculated using the two-dimensional Einstein–Stokes equation [10].

### 2.7. Atomic Force Microscopy

For atomic force microscopy (AFM), the vesicle samples were washed twice and resuspended in sodium phosphate buffer (pH 7.4). A drop of the suspension was placed on a coverslip and dried in a vacuum oven for 30 min. AFM was then performed using Dimension FastScan (Bruker, Billerica, MA, USA). AFM images were obtained in the PeakForceQNM mode (quantitative nanomechanical mapping) using standard silicon cantilevers ScanAsyst-Fluid+ (Bruker, Billerica, MA, USA) with a curvature of 2 nm and stiffness of 0.7 N/m. PeakForceQNM is a non-resonant, discontinuous contact method of AFM measurement. The probe moved vertically in a sinusoidal manner.

### 2.8. Total Protein Assay

MVs and cell pellets were lysed using a RIPA buffer (Thermo Fisher, Waltham, MA, USA) with the addition of Halt Protease Inhibitor Cocktail (Thermo Fisher, Waltham, MA, USA) to prevent protein degradation. Lysates were incubated for 30 min at 4 °C and then centrifuged at 16,000× *g* for 30 min at 4 °C. The supernatant was collected and aliquoted for storage at −80 °C. The total protein concentration in the samples was determined using a Pierce BCA Protein Assay Kit (Thermo Fisher, Waltham, MA, USA) according to the manufacturer’s instructions. Optical density (OD) measurements were performed at 562 nm using an Infinite 200 Pro microplate reader (Tecan, Männedorf, Switzerland).

### 2.9. Western Blot Analysis

For immunoblotting, samples were subjected to SDS-PAGE electrophoresis in TGX FastCast 12% gel (Bio-Rad, Hercules, CA, USA) at 70 V and 180 V. Briefly, 1 µg of total protein per well was loaded for primary T-cell samples, and 2 µg of protein per well for SupT1 cell samples. Western C protein ruler (Bio-Rad, Hercules, CA, USA) was used for size estimation. The SDS-PAGE gels were transferred to 0.45 µm PVDF membranes (Bio-Rad, Hercules, CA, USA) using a Mini Trans-Blot Cell (Bio-Rad, Hercules, CA, USA) at a constant voltage of 17 V for 30 min using Turbo Transfer Buffer (Bio-Rad, Hercules, CA, USA). The membrane was blocked with 5.0% nonfat dry milk in PBS-Tween for 1.5 h at room temperature. The β-actin antibody (A00730, GenScript, Piscataway, NJ, USA) was used at a dilution of 1:2000 in 5% milk-PBS-Tween for staining for 1.5 h at room temperature. The membranes were then washed with PBST-Tween for 4 × 15 min. The primary antibody to the protein of interest was used overnight at 4 °C at the recommended dilution with slow mixing. The membranes were washed with PBS-Tween for 4 × 15 min. HRP-conjugated secondary antibody was then added at the recommended dilution in 5% milk-PBS-Tween and incubated for 1 h at room temperature, followed by washing with PBS-Tween for 4 × 15 min. Then, 1.2 mL Pierce ECL Western Blotting Substrate (Bio-Rad, Hercules, CA, USA) was added, and the membranes were incubated for 2–5 min. Images of the membranes were obtained using the ChemiDoc XRS+ system (Bio-Rad, Hercules, CA, USA). The images were processed densitometrically in ImageJ (1.53k) software for relative protein amount estimation and β-actin normalization.

List of antibodies used: CD3 (ab699, Abcam, Waltham, MA, USA), granzyme B (ab134933, Abcam, Waltham, MA, USA), MHCII (ab180779, Abcam, Waltham, MA, USA), calnexin (MAA280Hu22, Cloud-Clone, Wuhan, China), HSP-70 (AF5466, Affinity, Darlinghurst, Australia), Lamin B1 (PAF548Mi01, Cloud-Clone, Wuhan, China), anti-rabbit HRP-conjugated antibody (ab9751, Abcam, Waltham, MA, USA), anti-mouse HRP-conjugated antibody (ab205729, Abcam, Waltham, MA, USA).

### 2.10. Fluorometric Assay

The fluorometric assay was performed using a VarioSkan microplate reader (Thermo Fisher, Waltham, MA, USA) in a black 96-well plate. For the DiI (Thermo Fisher, Waltham, MA, USA) and Calcein AM (Abcam, Waltham, MA, USA) assays, cells were pre-stained prior to obtaining vesicles. Cells were washed twice with DPBS, stained with the dyes for 20 min in a CO_2_ incubator, washed twice to remove the dyes, and then used for AV induction. A suspension of vesicles from stained cells in DPBS was used for the fluorometric assay. For Hoechst 33258 (Abcam, USA), AVs were first obtained and then stained with Hoechst 33258 for 20 min.

### 2.11. Flow Cytometry

For flow cytometry, 3 × 10^5^ cells were harvested, washed twice with DPBS, and stained with antibodies for 20 min. Cells were then washed twice to remove antibodies, and flow cytometry analysis was performed using FACS Aria III (BD Biosciences, Fremont, CA, USA). List of antibodies used: CD3-FITC (344804, Biolegend, San Diego, CA, USA), CD4-APC (354408, Biolegend, San Diego, CA, USA), and CD8-PE (344706, Biolegend, San Diego, CA, USA).

### 2.12. Statistical Analysis

Kruskal–Wallis test with Dunn’s post hoc test was used for atomic force microscopy data. One-way ANOVA with Tukey post hoc test was used for fluorometric and total protein data, except for the Hoechst33258 assay, for which a *t*-test was used. Data analysis was performed using GraphPad Prism 5. In the graphs, *p*-value < 0.05 is marked *, *p* < 0.01 is marked **, and *p* < 0.001 is marked ***.

## 3. Results

### 3.1. Cytochalasin B and Ultrasonication Result in T-Cell AVs with Different Yield, Size Distribution, and Mechanical Properties

Our aim was to investigate the ability of cytochalasin B (ChB) and ultrasonication (US) to induce the generation of AVs from T cells and to assess their yield and size. To achieve this, we first isolated PBMCs, which were then activated with CD3/CD28 antibodies and expanded for two weeks in media containing IL-2. The cells were then washed twice with DPBS, split in half, and placed in optimized serum-free media. One half was left to incubate, while the other was first reactivated with CD3/CD28 antibodies and then incubated. After 48 h, the supernatants were collected as a source of natural T cell-derived MVs. The cells were then washed twice with DPBS and used to obtain AVs. All samples were processed in parallel by differential centrifugation (see list of samples in Table 1 and scheme in Figure 2).

The obtained samples were analyzed using nanoparticle tracking analysis (Figure 3). The particle concentration was measured and used to calculate the microvesicle yield per parent cell (Figure 4A). In general, AVs generated with ChB or US showed a lower mean and mode size than natural MVs (Figure 4B). For natural MVs, the yield was equal to 1858 particles per cell, whereas after activation, this number increased three-fold. Cytochalasin B allowed us to obtain AVs with a yield comparable to that of natural MVs. Ultrasonication alone produced AVs at a level similar to activation (4978 AVs US per T cell vs. 5437 aMVs per activated T cell), and ultrasonication combined with activation led to a seven-fold increase in the yield of AVs compared to natural MVs. Using a similar ultrasonication induction method, Wang et al. reported an 18.5-fold increase in AV yield from mesenchymal stem cells (MSCs) compared to natural MVs [11]. Interestingly, cytochalasin B-induced AVs from renal carcinoma cell lines (786-O, ACHN) showed only a 1.8–3.7-fold increase in yield compared to natural MVs [12].

In terms of particle size distribution, we found that both chemical (ChB) and physical (US) induction methods resulted in more uniform MVs with lower dispersion and smaller average size than natural MVs in general (Figure 3). Functional analysis revealed interesting patterns, suggesting that irrespective of the type of induction, a population of vesicles with a size of 77 ± 3 nm was present in all samples. Another population of small vesicles 114–115 nm in size was also observed (Figure 3A,D). We assume that both 77 ± 3 nm and 114–115 nm particles represent exosomes that are characteristic of natural vesicles from resting and activated T cells. The next well-differentiated population of natural MVs is 169 ± 5 nm in size, while other larger fractions potentially can be attributed to smaller vesicle associates, as observed in a recent study [8]. Cytochalasin B-induced membrane vesicles demonstrated lower polydispersity with a significant population of small particles 51 ± 5 nm in size (Figure 3B,E). At the same time, vesicles generated with ChB from activated T cells contained many larger particle clusters, consistent with AFM data (Figure 5). Ultrasonication-induced vesicles showed a similar set of clusters but with the opposite correlation between resting and activated T cells; vesicles from activated T cells were predominantly represented by a 45–52 nm population with smaller mean size and polydispersity (Figure 3C,F). These observations were also consistent with AFM data (Figure 5C,F).

AFM was used to visualize the vesicles (Figure 5) as one of the recommended methods of single-particle study by MISEV2023 (Minimal Information for Studies of Extracellular Vesicles 2023). In general, the obtained images are consistent with the NTA results in terms of particle size and dispersion. The samples that were the most homogeneous and contained smaller particles were generated with cytochalasin B from resting T cells and with ultrasonication from activated T cells, while naturally secreted MVs from resting or activated T cells were more variable in size and larger in general. The mechanical properties of the vesicles, such as adhesion and deformation, were also evaluated by AFM (Figure 6). We found that vesicles from activated T cells had higher adhesion (Figure 6A) than vesicles from resting T cells, and this parameter also increased across the natural/ChB/US range. The AFM probe was made of silicon nitride and was negatively charged due to oxidation during the experiment, so this observed interaction may be electrostatic in nature [13]. The activation of T cells results in an increased presence of surface proteins such as T-cell receptor (TCR) or major histocompatibility complex (MHC) that, along with many other proteins, may contain positively charged motifs available for electrostatic interaction [14].

Despite significant differences in deformation properties observed in some of the samples (Figure 6B), there seems to be a minor tendency toward lower deformation in vesicles generated from activated T cells and across the natural/ChB/US range.

### 3.2. AVs from SupT1 Cells Contain Various Cellular Components of the Donor Cells

SupT1 cells were used to estimate the transfer of various cellular components into generated AVs using fluorometric assays and Western blot analysis. Several fluorescent probes were selected for the fluorometric assay, including Calcein AM cell-permeant dye used to determine cell viability, Katushka2S exogenous red fluorescent protein expressed in the cytoplasm, DiI lipophilic dye for membrane staining, and Hoechst 33342 for dsDNA staining. We also aimed to determine whether AVs were formed by the disruption of the cell membrane or by membrane budding. To achieve this, we collected AVs from an equal number of SupT1(Kat+) cells at different dilutions (10 × 10^6^ cells/mL, 5 × 10^6^ cells/mL, and 2.5 × 10^6^ cells/mL) for samples stained with Calcein AM and DiI.

The total protein concentration measured in AVs by bicinchoninic acid assay confirmed that the yield was higher after US treatment compared to ChB, which is consistent with the NTA results. Furthermore, the yield was not affected by cell dilution for ChB-treated samples (Figure 7F and Figure 8C,F). The differences between diluted US-treated samples can be explained by the variation in sample volume, which is critical due to the physical basis of the method.

Initially, SupT1 cells were simultaneously stained with DiI and Calcein AM, and then AVs were generated by ChB or US treatment. DiI fluorescence showed a much higher yield of the lipid component for AVs US at higher dilutions, whereas for Avs ChB, the maximum yield was in the middle of the concentration range (Figure 7A,D). By contrast, no significant difference in fluorescence was observed across samples stained with Calcein AM (Figure 7B). Interestingly, when the fluorescence of Calcein AM-stained cells was normalized to total protein or lipid (DiI fluorescence) levels, we observed a tendency for a decreased normalized Calcein AM signal for highly diluted samples (Figure 7C,E). This may indicate membrane disruption during AVs US formation. In addition, for Avs US the Calcein AM fluorescence normalized to total protein content was much lower than for AVs ChB, suggesting a greater loss of cytoplasmic content.

From this point of view, the results for AVs obtained from SupT1(Kat+) cells may be somewhat ambiguous (Figure 8A–C). For both AVs ChB and AVs US an increase in Katushka2S fluorescence was observed, as the source cells were diluted. However, AVs US consistently showed lower Katushka2S fluorescence compared to AVs ChB, similar to that of Calcein AM staining.

Next, we determined whether the AVs contained residual parent cell DNA. To investigate this, AVs generated from SupT1 cells were stained with Hoechst 33258. The fluorometric assay showed that only AVs US, but not AVs ChB, contained dsDNA (Figure 8D–F). These results suggest that the method of AV induction can have a significant impact on the quality and cargo content of the resulting vesicles. Therefore, careful consideration of the method used for AV generation is critical for reliable experimental results.

Western blot (WB) analysis also confirmed the transfer of various cellular components from the parental cells to AVs (Figure 9). All samples were normalized to total protein before loading; however, they had different β-actin contents, most likely due to the discrepancies in transfer efficiency during induction. Therefore, non-normalized data are also provided for diligent analysis.

Calnexin (90 kDa) is an important membrane-bound component of the endoplasmic reticulum (ER) [15] and is recommended by MISEV2023 (Minimal Information for Studies of Extracellular Vesicles 2023) guidelines as a marker of ER content in EVs. WB analysis showed that AVs ChB contained ER amounts comparable to that of the parental SupT1 cells when normalized to β-actin, whereas AVs US contained much higher ER amounts. However, in absolute amounts (not normalized to β-actin), AVs ChB and AVs US demonstrated comparable ER levels, which were higher than in SupT1 cells (Figure 9A).

Hsp70 (70 kDa) is a chaperone protein found in several cellular compartments, including the nucleus and mitochondria [16]. However, according to the MISEV2023 guidelines, Hsp70 should be considered a cytosolic protein found inside EVs. Our results showed that EVs ChB contained less Hsp70 than parental SupT1 cells after normalization to β-actin. However, this may be due to the higher contents of β-actin in total protein rather than actual differences in Hsp70 levels. According to WB results, AVs US appeared to contain insignificant Hsp70 amounts, and it was difficult to distinguish this from the artifact signals of the non-specific binding (Figure 9B).

Lamin B1 is a key component of the nuclear lamina, which comprises the inner nuclear membrane and nucleoplasm [17]. The lack of Lamin B1 in AVs ChB suggests the absence of the nuclear fraction, consistent with the presumed induction mechanism (Figure 9C). Meanwhile, AVs US contained nuclear components from the parental cells, albeit at lower levels than in the cells themselves. Further investigation is required to determine the exact mechanism underlying the transfer of nuclear components in AVs US and whether this is due to the cell isolation process or a natural feature associated with their formation.

In summary, AVs ChB appear to be strikingly similar to their parental cells in terms of content, except for nuclear components. In contrast, AVs US were found to consist mainly of membranes and membrane-bound components of the parental cells. Further research is needed to fully understand the formation mechanism, composition, and functions of these types of AVs.

### 3.3. AVs from Primary T Cells Generated with Cytochalasin B or Ultrasonication Carry Key Functional Proteins

We also investigated whether T cell-derived extracellular vesicles contained functionally important proteins. The expression of CD3 on the surface of T cells is well known to become enhanced upon activation [18]. We aimed to determine whether CD3 is detectable on T cell-derived EVs. For that, we performed WB analysis and normalized the data to β-actin. Our results showed a relative increase in CD3 levels in all types of vesicles tested, especially in the case of AVs aUS obtained from activated T cells. Notably, natural MVs showed the highest absolute levels of CD3 (Figure 10A).

Granzyme B (30 kDa) is one of the major effectors of cytotoxic T lymphocyte (CTL)-mediated cell death that is released into the extracellular space upon T-cell activation, where it penetrates target tumor cells via perforin-mediated entry [19]. Our results showed that all vesicles obtained from activated T cells had similar levels of granzyme B (Figure 10B). The 60 kDa bands observed in the T-cell samples most likely represent dimers [20] that did not dissociate under SDS-PAGE conditions. However, no such dimers were found in the vesicle samples, except for AVs aChB.

HLA II expression reflects the antigen-presenting function of T lymphocytes and is predominantly known as a characteristic of CD4^+^ Th cells, which increases after activation [21]. HLA II is a heterodimer consisting of α-chain (approximately 35 kDa) and β-chain (approximately 25 kDa) that normally dissociates after boiling under SDS-PAGE conditions [22]. Interestingly, we found that the 35 kDa subunit was predominantly present in T-cell samples, whereas the 25 kDa subunit was much more abundant in microvesicle samples (Figure 10C). Nevertheless, the overall HLA II content in microvesicles appeared to be high, especially in the case of microvesicles generated from pre-activated T cells.

## 4. Discussion

The relatively well-established field of natural EVs is now being complemented by industry-oriented approaches to generate artificial vesicles with uniform properties and high yields. In this study, we aimed to evaluate the induction of T-cell microvesicles using cytochalasin B and ultrasonication and to compare the resulting products with natural EVs and parental cells.

We have shown that both cytochalasin B and ultrasonication can induce the formation of artificial vesicles (AVs) (Figure 11). Considering the number of particles and the total protein content, ultrasonication appears to be more efficient in producing higher quantities of AVs. However, in terms of the quality of the resulting AVs, these two induction methods yielded different results. In contrast, AVs induced by ultrasonication showed higher self-inclusion of membrane-bound proteins and parental cell DNA and had lower content of cytosolic and cytoplasmic proteins. Importantly, membrane disruption during ultrasonication could potentially be exploited to enhance AVs with therapeutic or otherwise functional payload, e.g., small-molecule drugs, genetic vectors, proteins, fluorescent compounds, etc.

In the biopharmaceutical industry, the manufacturing of AVs requires high efficiency, scalability, simplicity of technology, and a minimal chemical footprint. These features are necessary to make upstream and downstream processes cost-effective while ensuring that the final product is functional and safe. From this perspective, ultrasonication as an induction method appears to be one of the most attractive techniques for producing AVs for therapeutic purposes.

Cell-free therapy modality in the context of T cells is a highly promising technology. Nowadays, T cells are used and studied for the treatment of various diseases such as oncology, autoimmune disorders, and many others. In all these fields, artificial vesicles could contribute to significant progress, for example, by facilitating access through dense solid tumor microenvironment, overcoming immunosuppressive tumor resistance to CAR-T therapy, providing increased safety, and presenting the possibility of allogeneic production.

However, significant research gaps remain to be addressed by the scientific community. These include the need to investigate the content of AVs as a time- and cell-type dependent process and its impact on the properties of the resulting AVs, especially for activated immune cells as a source. In addition, the functional capacity of AVs is an open question that requires further investigation. The wide range of technology implementation gives rise to a number of promising research directions. Different cell sources, genetic modification of parent cells (e.g., to achieve allogeneicity or to increase specificity or efficacy), and additional loading with small molecules are promising but not limited avenues for further development.

## Figures and Tables

**Figure 1 biomedicines-12-00919-f001:**
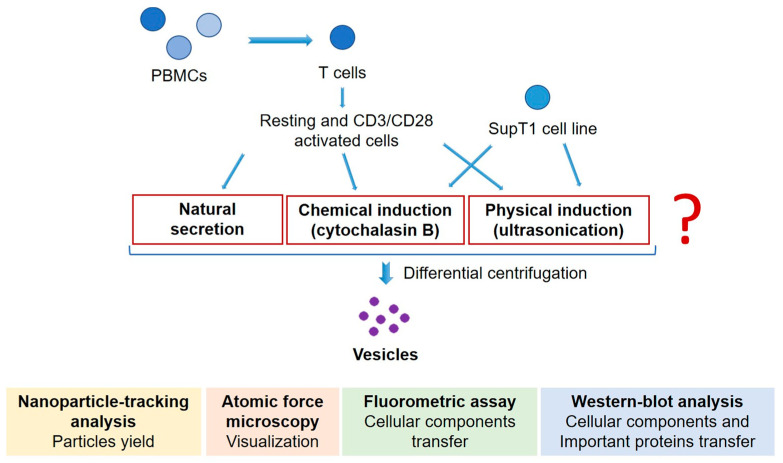
The scheme of the experiment includes the induction of artificial vesicles from primary T lymphocytes and SupT1 cell line with two methods under investigation and further comprehensive study of obtained vesicles.

**Figure 2 biomedicines-12-00919-f002:**
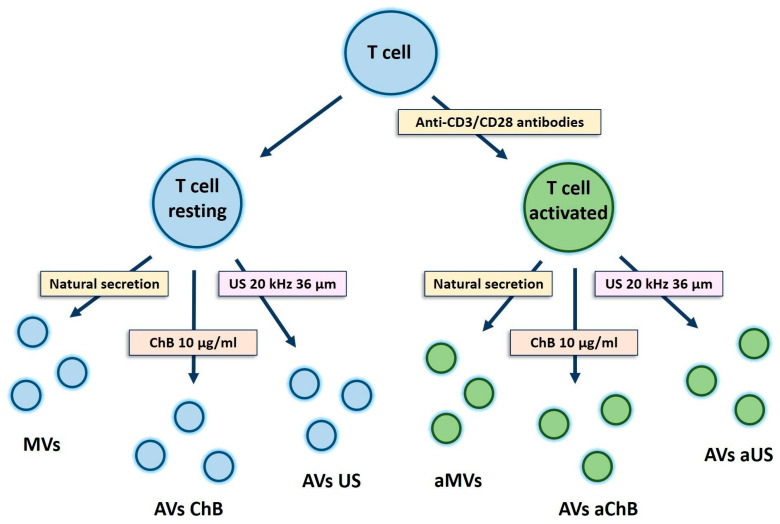
Scheme of experiments with primary T cells. Two types of cell sources were used—resting and activated T cells. Additionally, two types of induction were tested—chemical induction with cytochalasin B and physical induction with ultrasonication. Naturally secreted vesicles were used as a control.

**Figure 3 biomedicines-12-00919-f003:**
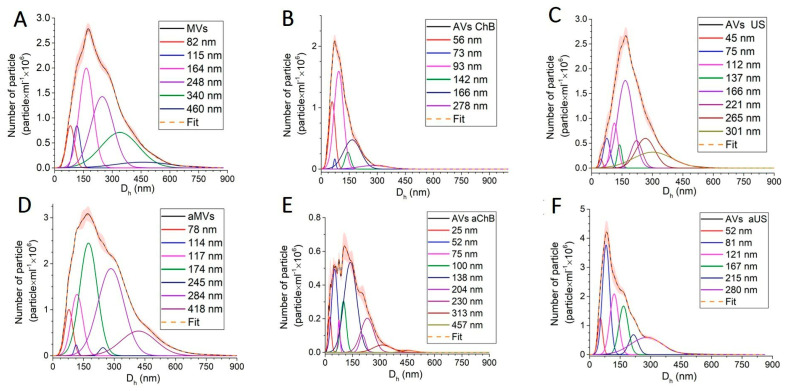
Size distribution measured by nanoparticle tracking analysis of vesicle samples generated from T cells: (**A**) MVs (naturally secreted from resting T cells); (**B**) AVs ChB (artificial vesicles generated with cytochalasin B from resting T cells); (**C**) AVs US (artificial vesicles generated with ultrasonication from resting T cells); (**D**) aMVs (MVs secreted by activated T cells); (**E**) AVs aChB (artificial vesicles generated with cytochalasin B from activated T cells); (**F**) AVs aUS (artificial vesicles generated with ultrasonication from activated T cells).

**Figure 4 biomedicines-12-00919-f004:**
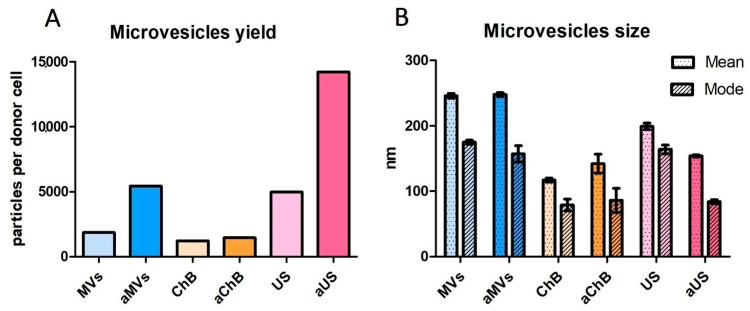
Comparison of yields and sizes of microvesicles generated using different induction techniques: (**A**) calculated microvesicle yields (number) per donor primary T cell (calculation is based on concentration measured by nanoparticle tracking analysis); (**B**) mean and mode microvesicle sizes measured by nanoparticle tracking analysis. MVs (naturally secreted from resting T cells); AVs ChB (artificial vesicles generated with cytochalasin B from resting T cells); AVs US (artificial vesicles generated with ultrasonication from resting T cells); aMVs (MVs secreted by activated T cells); AVs aChB (artificial vesicles generated with cytochalasin B from activated T cells); AVs aUS (artificial vesicles generated with ultrasonication from activated T cells). All samples were analyzed in five repetitions.

**Figure 5 biomedicines-12-00919-f005:**
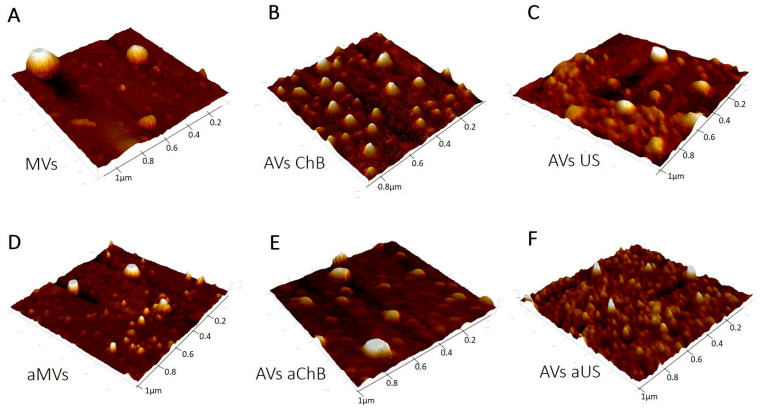
Images of microvesicle samples obtained by atomic force microscopy: (**A**) MVs (naturally secreted from resting T cells); (**B**) AVs ChB (artificial vesicles generated with cytochalasin B from resting T cells); (**C**) AVs US (artificial vesicles generated with ultrasonication from resting T cells); (**D**) aMVs (MVs secreted by activated T cells); (**E**) AVs aChB (artificial vesicles generated with cytochalasin B from activated T cells); (**F**) AVs aUS (artificial vesicles generated with ultrasonication from activated T cells).

**Figure 6 biomedicines-12-00919-f006:**
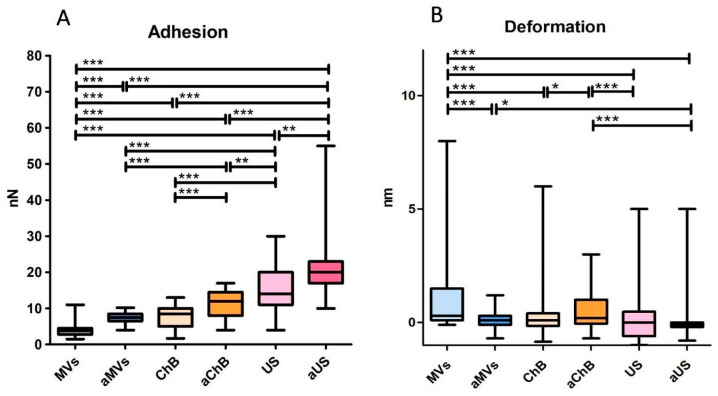
Adhesion and deformation properties of various microvesicles obtained by atomic force microscopy: (**A**) relative adhesion of microvesicles; (**B**) relative deformation of microvesicles, *n* = 85–220 depending on the sample. MVs (naturally secreted from resting T cells); AVs ChB (artificial vesicles generated with cytochalasin B from resting T cells); AVs US (artificial vesicles generated with ultrasonication from resting T cells); aMVs (MVs secreted by activated T cells); AVs aChB (artificial vesicles generated with cytochalasin B from activated T cells); AVs aUS (artificial vesicles generated with ultrasonication from activated T cells). *p*-value < 0.05 is marked *, *p* < 0.01 is marked **, and *p* < 0.001 is marked ***.

**Figure 7 biomedicines-12-00919-f007:**
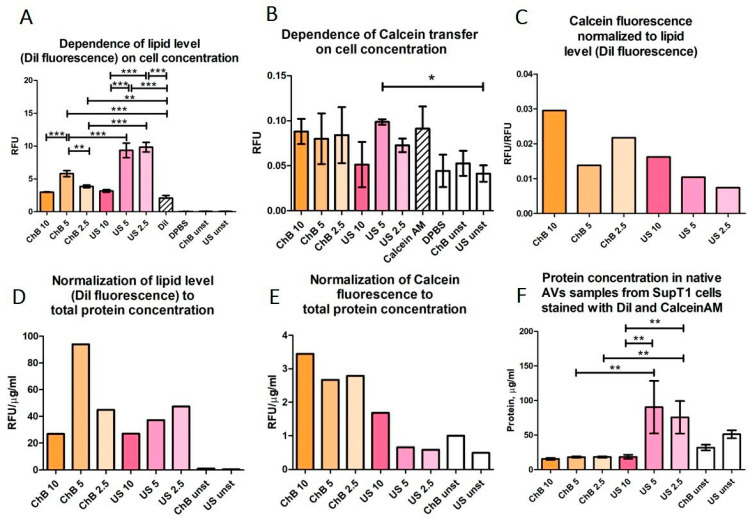
Transfer of SupT1 cytoplasmic compartments to AVs measured by fluorometric assay. ChB 10, ChB 5, and ChB 2.5 refer to AV samples obtained by cytochalasin B induction of cell suspensions diluted to 10 × 10^6^ cells/mL, 5 × 10^6^ cells/mL, or 2.5 × 10^6^ cells/mL, respectively. US 10, US 5, US 2.5 refer to AV samples obtained by ultrasonication induction of cell suspensions diluted to 10 × 10^6^ cells/mL, 5 × 10^6^ cells/mL, or 2.5 × 10^6^ cells/mL, respectively. All other samples were prepared from 5 × 10^6^ cells/mL SupT1 suspension. Unst—vesicles obtained from unstained cells; DPBS—clear DPBS; DiI or Calcein—DPBS with dye at concentration used for staining: (**A**) fluorescence of DiI in samples obtained from different cell dilutions; (**B**) fluorescence of Calcein in samples obtained from different cell dilutions; (**C**) Calcein fluorescence normalized to lipid level; (**D**) lipid level normalized to total protein concentration; (**E**) Calcein fluorescence normalized to total protein concentration; (**F**) protein concentration in native samples obtained from different cell dilutions. All samples were analyzed in triplets. *p*-value < 0.05 is marked *, *p* < 0.01 is marked **, and *p* < 0.001 is marked ***.

**Figure 8 biomedicines-12-00919-f008:**
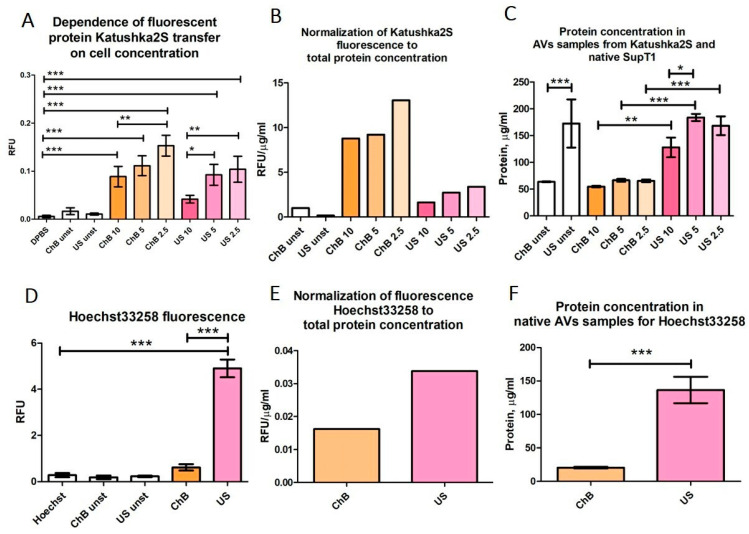
Transfer of SupT1 cytoplasmic fluorescent protein Katushka2S and nuclei to AVs measured by fluorometric assay: (**A**–**C**) Fluorometric data of Katushka2S transfer into AVs; ChB 10, ChB 5, and ChB 2.5 refer to AV samples induced with cytochalasin B from cell suspension diluted to 10 × 10^6^ cells/mL, 5 × 10^6^ cells/mL, or 2.5 × 10^6^ cells/mL, respectively. US 10, US 5, and US 2.5 refer to AV samples induced with ultrasonication from cell suspension diluted to 10 × 10^6^ cells/mL, 5 × 10^6^ cells/mL, or 2.5 × 10^6^ cells/mL, respectively. All other samples were obtained from SupT1 cells at 5 × 10^6^ cells/mL. (**D**–**F**) Fluorometric data of AVs, stained by Hoechst33258 for detection of double-strand DNA. DPBS—clear DPBS; ChB unst, US unst—samples of vesicles, obtained from non-fluorescent or unstained cells; Hoechst—DPBS with dye at concentration used for staining. All samples were analyzed in triplets. *p*-value < 0.05 is marked *, *p* < 0.01 is marked **, and *p* < 0.001 is marked ***.

**Figure 9 biomedicines-12-00919-f009:**
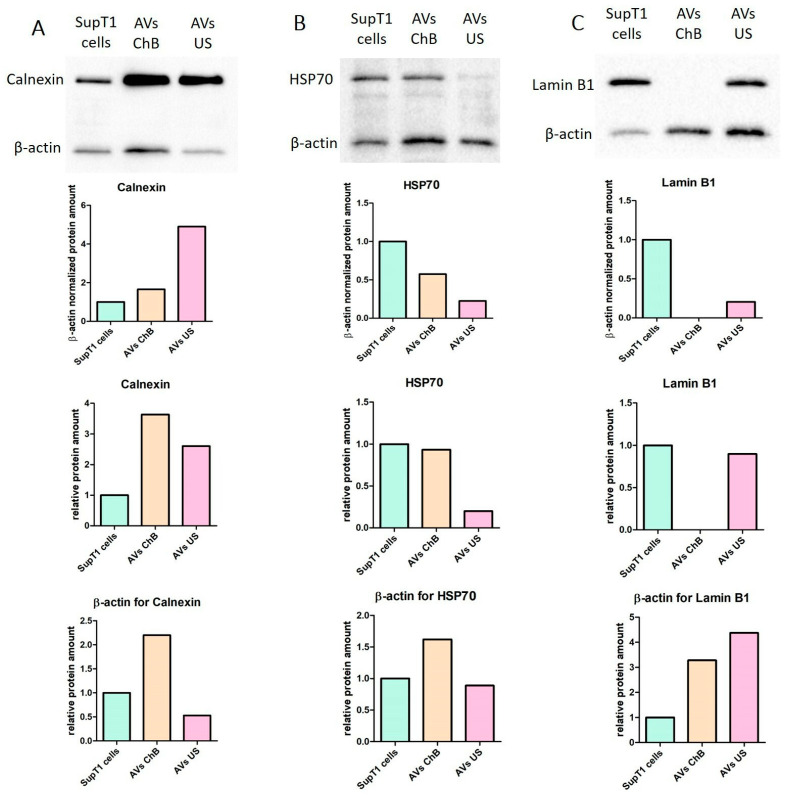
Transfer of SupT1 cell compartment into AVs measured by immunoblotting analysis: (**A**) immunoblotting analysis of calnexin (endoplasmic reticulum marker); (**B**) immunoblotting analysis of Hsp70 (cytosolic marker); (**C**) immunoblotting analysis of lamin B1 (nuclear marker). Data include immunoblotting images, calculated levels of the normalized target protein, and β-actin.

**Figure 10 biomedicines-12-00919-f010:**
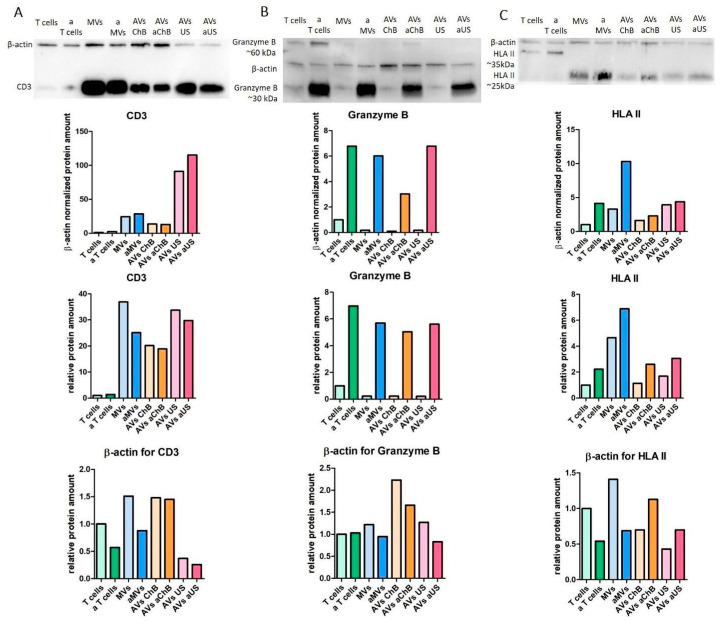
Functionally important proteins (CD3, granzyme B, and HLA II) presented in different types of AVs, obtained from resting T cells (MVs, AVs ChB, and AVs US) or activated T cells (aMVs, AVs aChB, and AVs aUS): (**A**) immunoblotting analysis of CD3; (**B**) immunoblotting analysis of granzyme B; (**C**) immunoblotting analysis of HLA II. Data include immunoblotting images, calculated levels of the normalized target protein, and β-actin.

**Figure 11 biomedicines-12-00919-f011:**
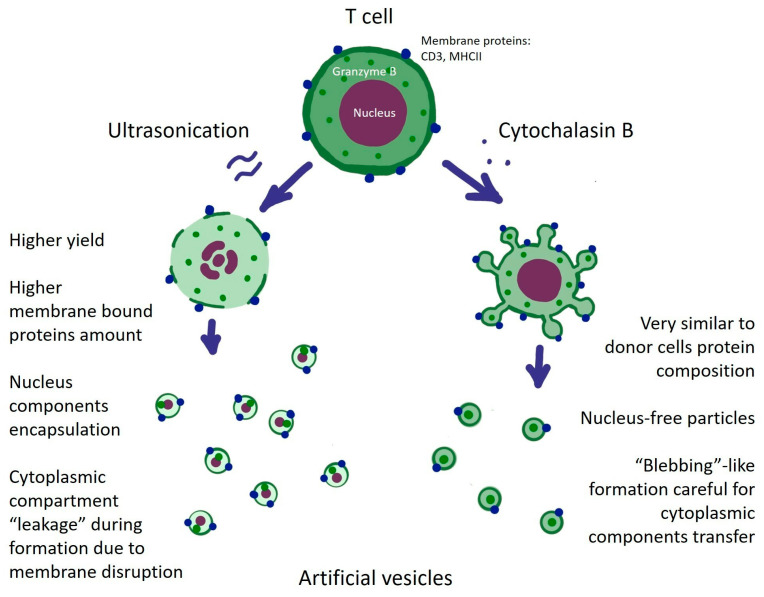
Formation of artificial vesicles from T cells using physical induction by ultrasonication and chemical induction by cytochalasin B. Ultrasonication causes disruption of the cell membrane with partial leakage of cell contents, while membrane proteins are transferred to vesicles. Cytochalasin B induces vesiculation from the cell surface with effective intracellular transfer of cytoplasmic contents. In both cases, functional components such as granzyme B are transferred to the resulting vesicles.

**Table 1 biomedicines-12-00919-t001:** List of sample abbreviations and methods of induction.

Cell Source	Method of Induction	Sample Abbreviation
Resting T cells	No (natural secretion)	MVs (naturally secreted microvesicles)
Resting T cells (or SupT1 cells)	Cytochalasin B at 10 µg/mL for 30 min, with vortex	AVs ChB (artificial vesicles generated with cytochalasin B)
Resting T cells (or SupT1 cells)	Ultrasonication 20 kHz, amplitude 36 µm, 1 min (0.5 s sonication, 0.5 s pause)	AVs US (artificial vesicles generated with ultrasonication)
Activated T cells	No (natural secretion)	aMVs (natural microvesicles from activated T cells)
Activated T cells	Cytochalasin B at 10 µg/mL for 30 min, with vortex	AVs aChB (artificial vesicles from activated T cells generated with cytochalasin B)
Activated T cells	Ultrasonication 20 kHz, amplitude 36 µm, 1 min (0.5 s sonication, 0.5 s pause)	AVs aUS (artificial vesicles generated from activated T cells with ultrasonication)

## Data Availability

The data that support the findings of this study are not openly available and are available from the corresponding author upon reasonable request.

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
