# Peer review of "Artificial Extracellular Vesicles Generated from T Cells Using Different Induction Techniques"

_biomedicines, 2024, doi:10.3390/biomedicines12040919_

Round 1

Reviewer 1 Report

Comments and Suggestions for Authors

The authors prepared extracellular vesicles from T-cells. I have a few comments/questions have to fix.

1)      I would like to ask about the novelty of this work since the only preparation of EVs is not a novelty. The generation of EVs has been carried out by previous research a couple of times. In this case, the authors may need to change the title of their MS.

2)       The abstract should be rewritten completely. The current abstract is only general facts. In the revised version, the authors should write the important results, focusing on numerical results.

3)      I would like to recommend drawing an appropriate graphical abstract at the end of the introduction.

4)      Some figures do not have error bars. Like Figs 8, 9, 6C-E.

5)      Normal EVs usually contain microRNA inside and multiple markers on the surface. How the authors are sure the cells-induced cytochalasin B and ultrasonics produce similar EV profiles to untreated cells? Did the authors profile and compare these two groups? ( treated and untreated)  

6)      The authors did not describe the purpose of using AFM on vesicles. Moreover, they could calculate the Young modulus in a table and compare the obtained EVS and AVs in terms of mechanical properties. 

Author Response

We thank the Reviewer for time and effort, as well as valuable comments. Below we provide point-by-point replies.

The authors prepared extracellular vesicles from T-cells. I have a few comments/questions have to fix.

  • I would like to ask about the novelty of this work since the only preparation of EVs is not a novelty. The generation of EVs has been carried out by previous research a couple of times. In this case, the authors may need to change the title of their MS.

Answer: The aim of the current research was not focused on vesicles preparation in general, but rather on approbation and comparison of different techniques (such as ultrasonication or cytochalasin B) for vesicles induction to achieve high yields of T cell artificial vesicles. To the best of our knowledge, no major studies that apply these approaches to T lymphocytes have been published to date despite the fact that naturally secreted by T-cells EVs is a well-known topic. Hence the novelty of the study. From this point of view the title «Artificial Extracellular Vesicles Generated from T Cells Using Different Induction Techniques» seems to be correct.

  • The abstract should be rewritten completely. The current abstract is only general facts. In the revised version, the authors should write the important results, focusing on numerical results.

Answer: Done

  • I would like to recommend drawing an appropriate graphical abstract at the end of the introduction.

Answer: Done (Figure 1)

  • Some figures do not have error bars. Like Figs 8, 9, 6C-E.

Answer: Figures which without error bars contain normalized data (means ratio) from different methods of measurement where error bars can’t be calculated mathematically correct (Fig. 4A, Fig. 7C-E, Fig. 8B,E) or western-blot data which doesn’t have replicates (Fig. 9, 10).

  • Normal EVs usually contain microRNA inside and multiple markers on the surface. How the authors are sure the cells-induced cytochalasin B and ultrasonics produce similar EV profiles to untreated cells? Did the authors profile and compare these two groups? (treated and untreated)

Answer: Content and surface markers of naturally secreted EVs plays the main role in their functional activity. Protein and RNA profile of naturally secreted EVs and artificial-induced vesicles for sure would have significant differences which shall be thoroughly investigated in our future studies.  We thank Reviewer for this valuable comment.

  • The authors did not describe the purpose of using AFM on vesicles. Moreover, they could calculate the Young modulus in a table and compare the obtained EVS and AVs in terms of mechanical properties.

Answer: AFM is one of the techniques recommended by MISEV2023 (Minimal information for studies of extracellular vesicles: From basic to advanced approaches, DOI: 10.1002/jev2.12404) for single-particle visualization and morphology study. Mechanical properties are represented as deformation raw data (Fig. 6B) for comparison.

Reviewer 2 Report

Comments and Suggestions for Authors

Cell therapy is at the forefront of biomedicine in oncology and regenerative medicine. One of the major obstacles in the translation of EVs into practice is their low yield of production, which is insufficient to achieve therapeutic amounts. The authors evaluated two primary approaches of artificial extracellular vesicle induction in primary T cells and SupT1 cell line -cytochalasin B as a chemical inducer and ultrasonication as a physical inducer. They found that both methods are capable of producing artificial vesicles. The most effective approach for T-cell induction in terms of number of vesicles seems to be the combination of anti-CD3/CD28 antibody activation with ultrasonication.

Major concern

The different stimulation by pharmacologic agents or physical stress will regulate the expressions of proteins or miRNA in the EVs by T cells. The changes of the contents in the EVs active different signal pathways and responses in receipt cells. The authors did not evaluate and compare the changes of proteins or miRNA expressions in these EVs.

The topic of this study is interesting. However, the results of this study are superficial, further research is needed.

Minor concern

Line 48. “CD8+” should be “CD8+”.

Line 82, 98. “1*106 cells/ml” should be “1 x 106 cells/ml”.

Line 100, 109, 182. “CO2” should be “CO2”.

Line 187. “3*105 cells” should be “3 x 105 cells”

Line 316, 329. “10*106 cells/ml” should be “1 x 107”.

Line 316, 329. “5*106 cells/ml” should be “5 x 106”.

Line 317, 328, 329. “2.5*106” should be “2.5 x 106”.

Comments on the Quality of English Language

Minor editing of English language required.

Author Response

We thank the Reviewer for time and effort, as well as valuable comments. Below we provide point-by-point replies.

  • Cell therapy is at the forefront of biomedicine in oncology and regenerative medicine. One of the major obstacles in the translation of EVs into practice is their low yield of production, which is insufficient to achieve therapeutic amounts. The authors evaluated two primary approaches of artificial extracellular vesicle induction in primary T cells and SupT1 cell line -cytochalasin B as a chemical inducer and ultrasonication as a physical inducer. They found that both methods are capable of producing artificial vesicles. The most effective approach for T-cell induction in terms of number of vesicles seems to be the combination of anti-CD3/CD28 antibody activation with ultrasonication.

Major concern

The different stimulation by pharmacologic agents or physical stress will regulate the expressions of proteins or miRNA in the EVs by T cells. The changes of the contents in the EVs active different signal pathways and responses in receipt cells. The authors did not evaluate and compare the changes of proteins or miRNA expressions in these EVs.

Answer: Content and surface markers of naturally secreted EVs plays the main role in their functional activity. Protein and RNA profile of naturally secreted EVs and artificial-induced vesicles for sure would have significant differences due to different mechanism of formation, but since the time of induction is very short (1 min for sonication and 30 min for cytochalasin B), probably it’s not enough to cause change in expression levels. Nevertheless, it’s an open question for further investigation. We thank Reviewer for this valuable comment.

  • The topic of this study is interesting. However, the results of this study are superficial, further research is needed.

Minor concern

Line 48. “CD8+” should be “CD8+”.

Line 82, 98. “1*106 cells/ml” should be “1 x 106 cells/ml”.

Line 100, 109, 182. “CO2” should be “CO2”.

Line 187. “3*105 cells” should be “3 x 105 cells”

Line 316, 329. “10*106 cells/ml” should be “1 x 107”.

Line 316, 329. “5*106 cells/ml” should be “5 x 106”.

Line 317, 328, 329. “2.5*106” should be “2.5 x 106”.

Answer: Done

Comments on the Quality of English Language

Minor editing of English language required.

Answer: The manuscript has been checked by the native speaker.

Reviewer 3 Report

Comments and Suggestions for Authors

The article is devoted to the isolation and analysis of microvesicles in various ways and the determination of the most optimal method for obtaining microvesicles. Analysis of the contents of the vesicles for residual nuclear material and lipids, which was carried out by the authors, is of particular interest.

The authors indicate at the beginning of the article (line 70) that one of the methods of obtaining of vesicles would be physical stimulation (shear stress), but they do not describe this method in detail.  

There is no conclusion in the manuscript, which makes it difficult to understand exactly what conclusions were drawn by the authors.

In addition, in the literature review authors should add references devoted to the visualization of microvesicles and justify the choice of this particular (AFM) method for visualizing cellular vesicles.

I recommend checking all references, since reference 12 does not correspond to the citation in the text.

Author Response

We thank the Reviewer for time and effort, as well as valuable comments. Below we provide point-by-point replies.

  • The article is devoted to the isolation and analysis of microvesicles in various ways and the determination of the most optimal method for obtaining microvesicles. Analysis of the contents of the vesicles for residual nuclear material and lipids, which was carried out by the authors, is of particular interest.

The authors indicate at the beginning of the article (line 70) that one of the methods of obtaining of vesicles would be physical stimulation (shear stress), but they do not describe this method in detail.

Answer: To avoid misunderstanding we now deleted this part. Physical stimulation is a component of cytochalasin B induction method implying gentle vortexing after incubation period (described in materials and methods section).

  • There is no conclusion in the manuscript, which makes it difficult to understand exactly what conclusions were drawn by the authors.

Answer: Conclusions for each data are provided in the Results section. General consideration and concluding remarks are located in the Discussion section.

  • In addition, in the literature review authors should add references devoted to the visualization of microvesicles and justify the choice of this particular (AFM) method for visualizing cellular vesicles.

Answer: AFM is one of the methods recommended by MISEV2023 (Minimal information for studies of extracellular vesicles: From basic to advanced approaches, DOI: 10.1002/jev2.12404) for single-particle visualization and morphology study. Reference is added to the manuscript.

  • I recommend checking all references, since reference 12 does not correspond to the citation in the text.

Answer: Corrected

Round 2

Reviewer 1 Report

Comments and Suggestions for Authors

The authors performed the required edition. 

Reviewer 2 Report

Comments and Suggestions for Authors

Gomzikova et al., reported the immunosuppressive properties of cytochalasin B-induced membrane vesicles of mesenchymal stem cells. They compared the immunomodulatory effect of cytochalasin B-induced microvesicles (CIMVs) and natural EVs from mesenchymal stem cells, and they observed no immunosuppression in mice pretreated with natural EVs, whereas CIMVs suppressed antibody production in vivo. Therefore, cytochalasin B may regulate the expressions of proteins or miRNA in the EVs derived from T cells in this study. Although the authors revised the manuscript, they failed to response my major concern. The authors should address the changes of proteins or miRNA expressions in these EVs in the manuscript.

Gomzikova MO, Aimaletdinov AM, Bondar OV, Starostina IG, Gorshkova NV, Neustroeva OA, Kletukhina SK, Kurbangaleeva SV, Vorobev VV, Garanina EE, Persson JL, Jeyapalan J, Mongan NP, Khaiboullina SF, Rizvanov AA. Immunosuppressive properties of cytochalasin B-induced membrane vesicles of mesenchymal stem cells: comparing with extracellular vesicles derived from mesenchymal stem cells. Sci Rep. 2020 Jul 1;10(1):10740. doi: 10.1038/s41598-020-67563-9.